 

# Immunocompetent mouse model for Crimean-Congo hemorrhagic fever virus

David W Hawman[1]\*, Kimberly Meade-White[1], Shanna Leventhal[1], Friederike Feldmann[1], Atsushi Okumura[1], Brian Smith[2], Dana Scott[3], Heinz Feldmann[1]\*

[1]Laboratory of Virology, Division of Intramural Research, NIAID, NIH, Hamilton, United States; [2]Texas Veterinary Pathology, Spring Branch, United States; [3]Rocky Mountain Veterinary Branch, Division of Intramural Research, NIAID, NIH, Hamilton, United States

**Abstract** Crimean-Congo hemorrhagic fever (CCHF) is a severe tick-borne febrile illness with wide geographic distribution. CCHF is caused by infection with the Crimean-Congo hemorrhagic fever virus (CCHFV) and case fatality rates can be as high as 30%. Despite causing severe disease in humans, our understanding of the host and viral determinants of CCHFV pathogenesis are limited. A major limitation in the investigation of CCHF has been the lack of suitable small animal models. Wild-type mice are resistant to clinical isolates of CCHFV and consequently, mice must be deficient in type I interferon responses to study the more severe aspects of CCHFV. We report here a mouse-adapted variant of CCHFV that recapitulates in adult, immunocompetent mice the severe CCHF observed in humans. This mouse-adapted variant of CCHFV significantly improves our ability to study host and viral determinants of CCHFV-induced disease in a highly tractable mouse model.

## Introduction

Crimean-Congo hemorrhagic fever virus (CCHFV) is the cause of Crimean-Congo hemorrhagic fever (CCHF). CCHFV is among the most widely distributed hemorrhagic fever viruses with cases reported through Africa, the Middle East, Asia, and Southern and Eastern Europe (*Bente et al., 2013*). Ticks of the *Hyalomma* genus are the principal vector and reservoir for CCHFV and cases of CCHF closely follow the geographic range of *Hyalomma* ticks (*Bente et al., 2013*). Climate change is leading to expansion of the range for *Hyalomma* ticks and recently *Hyalomma* ticks were found as far north as Sweden (*Grandi et al., 2020*). CCHF begins as a non-specific febrile illness that can rapidly progress to hemorrhagic disease (*Ergönül, 2006*), and there are currently no widely approved vaccines nor antivirals for CCHF. Case fatality rates can be as high as 30% (*Bente et al., 2013*).

To date, mouse models of CCHF have been limited to mice deficient in type I IFN responses, either through genetic deficiency such as interferon alpha receptor knock-out (*Ifnar1*[-/-]) (*Zivcec et al., 2013*; *Bente et al., 2010*; *Bereczky et al., 2010*) or through transient deficiency by antibody-mediated blockade of the interferon alpha receptor (*Garrison et al., 2017*; *Lindquist et al., 2018*). Infection of these mice typically results in a rapid onset fatal disease with many similarities to fatal human cases (*Zivcec et al., 2013*; *Bente et al., 2010*), although our group has recently developed a model that recapitulates the convalescent phase of CCHF (*Hawman et al., 2019*). Nevertheless, the lack of type I interferon in these models limits their usefulness for studying innate immunity to CCHFV, the rapid onset lethal disease in most of these models precludes study of later host responses and lack of type I interferon can impact adaptive immunity following infection or vaccination (*Clarke and Bradfute, 2020*).

We therefore sought to select a variant of CCHFV that was able to cause disease in fully immuno-competent mice. We serially passaged the clinical isolate, CCHFV strain Hoti, in mice deficient in

**\*For correspondence:**
david.hawman@nih.gov (DWH);
feldmannh@niaid.nih.gov (HF)

**Competing interests:** The authors declare that no competing interests exist.

adaptive immunity (recombination-activating-gene two deficient, $Rag2^{-/-}$) and wild-type (WT) C57BL/6J mice to generate a mouse-adapted variant of CCHFV (MA-CCHFV). In contrast to the parental CCHFV strain, MA-CCHFV was able to cause severe disease in WT mice that was associated with replication to high titers in multiple tissues, severe pathology in the liver and a severe inflammatory immune response. Unexpectedly, we identified a significant sex-linked bias in disease severity with female mice largely resistant to severe disease. In addition, we found that both host innate and adaptive immune responses are necessary to survive MA-CCHFV infection. Cumulatively, we report here a mouse-adapted variant of CCHFV that recapitulates in WT mice many aspects of severe human cases of CCHF.

## Results

### Mouse-adaptation of CCHFV strain Hoti

The ability of human clinical isolates of CCHFV to cause disease in type I IFN deficient but not sufficient mice (*Hawman et al., 2018*; *Hawman et al., 2019*; *Oestereich et al., 2014*; *Lindquist et al., 2018*) suggests CCHFV is unable to antagonize mouse innate immunity. We hypothesized that chronic infection and serial passage of CCHFV within the livers of $Rag2^{-/-}$ mice, which possess intact innate immune responses but lack adaptive immunity, would select for CCHFV variants that had adapted to overcome mouse innate restriction factors. This approach has successfully resulted in mouse-adaptation of the unrelated Zika and chikungunya viruses (*Hawman et al., 2017*; *Gorman et al., 2018*). We therefore infected a $Rag2^{-/-}$ mouse with the clinical isolate CCHFV strain Hoti by the intraperitoneal (IP) route and for the first passage collected blood at 4 weeks post-infection (WPI). For passage 2, we inoculated a naive $Rag2^{-/-}$ mouse with this blood and for remaining passages collected liver tissue when mice were exhibiting severe clinical signs of disease (hunched posture, piloerection, lethargy, weight loss). Liver tissue was homogenized, clarified of large debris by centrifugation and inoculated IP into naive $Rag2^{-/-}$ mice. At each passage, liver tissue from an individual mouse was passed into an individual naive mouse without purification or isolation . This serial passaging in $Rag2^{-/-}$ mice was performed nine times during which we observed a decrease in time of onset of severe disease (> day 28 post-infection (PI) for passage 1 to <day 7 PI passage 9) (*Figure 1—figure supplement 1A–B*). We performed a final two passages in the liver tissue of wild-type C57BL/6J mice for 11 total passages in mice. To monitor mouse adaptation during passaging, we evaluated inoculation of small groups of wild-type mice with virus stocks grown in tissue culture from homogenized liver tissue after $Rag2^{-/-}$ passage 4 (*Figure 1—figure supplement 1C*) and 9 (*Figure 1—figure supplement 1D*). As soon as passage 4, we observed transient weight loss in wild-type mice infected with passaged virus (*Figure 1—figure supplement 1C*). Weight loss after inoculation with later passages was associated with other clinical signs of disease such as piloerection, hunched posture and lethargy (data not shown) suggesting we had selected for a variant of CCHFV capable of causing severe disease in WT mice. After 11 total passages, we grew a virus stock in vitro on SW13 cells, hereafter termed MA-CCHFV. MA-CCHFV was sequenced by Illumina-based deep sequencing to exclude contamination and titered by SW13 median tissue culture infectious dose assay ($TCID_{50}$). Upon infection of WT mice, this variant caused substantial weight loss and clinical disease in male WT mice (*Figure 1—figure supplement 1E*). However, unexpectedly, infected female mice exhibited milder signs of disease (*Figure 1—figure supplement 1E*).

### MA-CCHFV causes severe disease in male C57BL/6J mice

To more fully characterize the clinical disease caused by MA-CCHFV, we infected 8-week-old WT C57BL/6J mice with 10,000 $TCID_{50}$ of MA-CCHFV via the IP route. For comparison, a group of mice were infected with an identical dose of parental strain CCHFV Hoti or mock infected. As expected, inoculation of WT mice with CCHFV strain Hoti resulted in no clinical disease besides transient weight loss on day 1 PI (<5%) (*Figure 1A,B*). In contrast, inoculation of male WT mice with MA-CCHFV resulted in severe clinical disease with significant weight loss beginning on day 4 PI and peaking on day 6 PI (*Figure 1A*). In addition to weight loss, male mice infected with MA-CCHFV exhibited overt clinical signs such as piloerection, hunched posture, and lethargy. Nearly all mice began to recover beginning on day 7 PI (*Figure 1A*). During our studies with MA-CCHFV in this report, lethal outcome in male mice was occasionally observed in some cohorts (1 of 32, 1 of 8, 2 of

**Figure 1.** MA-CCHFV causes overt clinical disease in wild-type (WT) mice. (A and B) Groups of 8-week-old male (A) or female (B) WT C57BL/6J mice were infected with 10,000 TCID$_{50}$ of MA-CCHFV or CCHFV Hoti via the intraperitoneal (IP) route and weighed daily. (A and B) Male and female mice were mock infected for comparison and same data is shown in both panels for comparison. N = 8 mock and four mice per CCHFV-infected group. Data shown as mean plus standard deviation. Statistical comparison performed using two-way ANOVA with Dunnett's multiple comparison to mock-infected mice. *p<0.05, ***p<0.001, ****p<0.0001. (C and D) Groups of male (C) or female (D) 8-week-old WT mice were infected with indicated dose of MA-CCHFV via the IP route and weighed daily. N = 5 mice per group. Studies were performed once.

The online version of this article includes the following source data and figure supplement(s) for figure 1:

**Source data 1.** Source data for Figure 1.
**Figure supplement 1.** Passage of Crimean-Congo hemorrhagic fever virus (CCHFV) in mice.
**Figure supplement 2.** Male versus female mice infected with MA-CCHFV.
**Figure supplement 3.** Seroconversion of dose-finding study.

12) indicating that lethal outcome is possible, albeit rare. Again, female WT C57BL/6J mice infected with MA-CCHFV exhibited a milder clinical disease with no significant weight loss compared to mock-infected mice (*Figure 1B*) and significantly less weight loss compared to male WT C57BL/6J mice infected with MA-CCHFV. (*Figure 1—figure supplement 2*). This was associated with milder clinical signs of disease.

We next infected mice IP with a range of doses of MA-CCHFV from 0.01 $TCID_{50}$ to 10,000 $TCID_{50}$ to determine the median infectious dose ($ID_{50}$) and to evaluate whether there was a correlation between virus dose and disease severity, as has been seen with mouse-adapted Ebola virus (*Haddock et al., 2018b*). Little-to-no clinical disease was observed in mice infected with 0.01 or 1 $TCID_{50}$ (*Figure 1C,D*). Male mice infected with 100 $TCID_{50}$ or greater showed weight loss (*Figure 1C*) and overt clinical signs of disease. Again, female mice infected with similar doses of MA-CCHFV exhibited milder weight loss compared to male mice (*Figure 1C,D*) that was also associated with milder overt clinical signs of disease. We evaluated sero-conversion to CCHFV at day 14 PI by whole-virion ELISA to confirm infection (*Figure 1—figure supplement 3*) and found that a dose of 0.01 $TCID_{50}$ resulted in productive infection of three of five male and four of five female mice. At doses of 1 $TCID_{50}$ and higher, all mice had detectable anti-CCHFV immunoglobulin at day 14 (*Figure 1—figure supplement 3*). Thus, the $ID_{50}$ of MA-CCHFV in WT male or female mice is <0.01 $TCID_{50}$. Cumulatively, these results demonstrated that doses of MA-CCHFV as low as 100 $TCID_{50}$ could cause disease in male WT C57BL/6J mice, although doses 10,000-fold lower still resulted in productive infection. For the rest of our studies, we infected mice IP with 10,000 $TCID_{50}$ of MA-CCHFV, unless otherwise indicated.

## MA-CCHFV causes disease in male mice of multiple laboratory strains

MA-CCHFV was generated by serial passage in mice on the C57BL/6J background. We wanted to determine if the MA-CCHFV phenotype was restricted to C57BL/6J mice or if MA-CCHFV could cause disease in other commonly used laboratory strains of mice. We therefore infected 8-week-old male and female C57BL/6J, C57BL6/NCr, 129S1, BALBc/J, or outbred CD1 mice IP with an intermediate dose of MA-CCHFV (1000 $TCID_{50}$). Similar to C57BL/6J mice (*Figure 2A*), male BALBc/J, C57BL6/NCr, and CD1 mice exhibited weight loss (*Figure 2C–E*) that was associated with overt clinical signs such as piloerection, hunched posture, and lethargy. Again, consistent with our data in C57BL/6J mice (*Figure 2A*), female mice of these strains exhibited milder clinical disease compared to the male mice (*Figure 2C–E*) demonstrating the sex-bias toward more severe disease in male mice is not restricted to the C57BL/6J strain. No mortality was observed in any of the mouse strains during this study. Interestingly, both male and female 129S1 mice appeared largely resistant to MA-CCHFV with mice exhibiting little-to-no weight loss (<5%) (*Figure 2B*) along with no overt signs of clinical disease. These data suggest that along with sex-linked differences there also exist genetic differences between mouse strains that result in distinct outcomes following infection with MA-CCHFV.

## MA-CCHFV causes lethal disease in young mice

Our data from adult (>8 weeks of age) WT C57BL/6J and several other commonly used laboratory strains of mice demonstrated that MA-CCHFV infection results in a severe but rarely fatal infection. A lethal model of MA-CCHFV infection would have utility for studies evaluating antiviral therapeutics or therapeutic interventions that seek to prevent CCHFV-induced mortality. For several viral infections, younger mice exhibit more severe disease than older mice (*Couderc et al., 2008*; *Johnson et al., 1972*) and neonatal but not adult WT mice are susceptible to non-adapted CCHFV infection (*Hoogstraal, 1979*). We hypothesized that young mice (3-week-old) mice may exhibit more severe disease upon infection with MA-CCHFV. Three-week-old male or female WT C57BL/6J mice were infected with 10,000 TCID50 of MA-CCHFV via the IP route. We found that infection of young male or female mice resulted in weight loss beginning on day 3 or day 4 (*Figure 2—figure supplement 1A*) and nearly all mice succumbed to the infection by day 7 PI (*Figure 2—figure supplement 1B*). Surviving male and female mice exhibited severe clinical disease but did not reach euthanasia criteria and began to rapidly recover after day 7 (*Figure 2—figure supplement 1A*). These data demonstrate that younger WT mice infected with MA-CCHFV are a suitable model for studying



**Figure 2.** MA-CCHFV causes clinical disease in multiple laboratory strains of mice. (A–E) Groups of 8-week-old male or female mice of indicated strains were infected with 1000 TCID$_{50}$ of MA-CCHFV via the intraperitoneal (IP) route and weighed daily. N = 5 mice per group. Data shown as mean plus standard deviation. Study was performed once.

The online version of this article includes the following source data and figure supplement(s) for figure 2:

**Source data 1.** Source data for Figure 2.

**Figure supplement 1.** Lethal infection of young mice with MA-CCHFV.

severe, lethal CCHF and that at younger ages, both male and female mice are similarly susceptible to severe disease.

## MA-CCHFV replicates to high titers in multiple tissues of adult WT mice

To determine if MA-CCHFV had an increased ability to replicate and disseminate in wild-type mice, we evaluated viral loads in several tissues of adult male and female WT mice infected with parental CCHFV strain Hoti or MA-CCHFV. We necropsied mice shortly after infection (1 DPI), early acute disease (3 DPI), peak clinical disease (6 DPI), early convalescence (8 DPI) and when mice had resolved all overt clinical signs of disease (14 DPI). In the plasma, at day 1 PI, mice infected with either Hoti or MA-CCHFV had similar RNA titers (p>0.05) (*Figure 3A*). In mice infected with CCHFV Hoti, viral RNA titers in the plasma rapidly declined after day 1 PI and continued to decline until they were near or below the limit of detection by day 8 PI indicating mice rapidly controlled the infection (*Figure 3A*). In contrast, viral RNA titers in the plasma of male mice infected with MA-CCHFV significantly increased between day 1 and day 3 PI (p<0.05) and these mice exhibited significantly greater



**Figure 3.** MA-CCHFV replicates to high titers of multiple tissues in wild-type (WT) mice. Groups of 8-week-old WT C57BL/6J mice were infected with 10,000 $TCID_{50}$ of MA-CCHFV or CCHFV Hoti via the intraperitoneal (IP) route. At indicated time points, mice were necropsied and viral RNA burdens in tissues evaluated by qRT-PCR. Statistical comparison performed with two-way ANOVA with Tukey's multiple comparison test. p-Values between MA-CCHFV and respective sex Hoti-infected mice indicated with * for females, # for males and between MA-CCHFV male and MA-CCHFV female mice with +. Plasma, liver, spleen, kidney, and lung: N = 2–4 (Hoti) and 7–8 (MA-CCHFV) per group per timepoint. Brain: N = 4 per group per timepoint. Study was performed once for Hoti and twice for MA-CCHFV. Data shown as mean plus standard deviation. Dashed line indicates limit of detection. *p<0.05, **p<0.01, ***p<0.001, ****p<0.0001.

The online version of this article includes the following source data for figure 3:

**Source data 1.** Source data for Figure 3.

viremia than Hoti-infected or female MA-CCHFV-infected mice at day 6 and day 8 PI (*Figure 3A*). Female mice infected with MA-CCHFV had similar titers to Hoti-infected mice (*Figure 3A*). Cumulatively, male mice infected with MA-CCHFV had significantly increased viremia compared to mice infected with CCHFV Hoti and consistent with more severe clinical disease, male mice infected with MA-CCHFV had higher and prolonged viremia compared to female mice.

We next evaluated viral RNA loads in the liver. At day 1 PI, male or female mice infected with parental strain Hoti or MA-CCHFV had similar viral RNA loads in the liver suggesting efficient dissemination to the liver independent of sex or virus strain (*Figure 3B*). However, viral RNA loads in mice infected with parental strain Hoti were similar at day 3 PI and began to decline at day 6 PI (*Figure 3B*) indicating these mice were able to efficiently control the non-adapted parental CCHFV strain Hoti. Viral loads in male mice infected with MA-CCHFV increased between day 1 and 6 PI (p<0.05) and did not begin to decline until day 8 PI (*Figure 3B*). Further, viral loads in these mice were significantly increased compared to male mice infected with CCHFV Hoti at days 3, 6, and 8 PI (*Figure 3B*). Female mice infected with MA-CCHFV had significantly elevated viral loads compared to female mice infected with CCHFV Hoti at day 3 PI (*Figure 3B*) but had similar viral loads to Hoti-infected mice thereafter. Consistent with more severe disease in male mice infected with MA-CCHFV, at day 6 and day 8 PI viral RNA loads in livers of male mice infected with MA-CCHFV were significantly greater than those in MA-CCHFV infected female mice (p<0.0001) (*Figure 3B*).

The spleen is another site of significant pathology and viral replication in CCHFV-infected *Ifnar1*$^{-/-}$ mice (*Hawman et al., 2019*) so we therefore evaluated viral loads in the spleen of mice infected with MA-CCHFV. Interestingly, at day 1 PI and at day 3 PI, viral loads were similar (p>0.05) between all groups and only at day 6 and day 8 PI did we see significantly increased burdens in the spleens of male mice infected with MA-CCHFV (*Figure 3C*). Thereafter, viral RNA loads declined in all groups, but viral RNA was still detectable in the spleens at day 14 PI (*Figure 3C*).

In addition to the plasma, liver, and spleen, we evaluated viral RNA loads in the kidneys, lungs, and brain, sites which we have previously seen high viral RNA loads in *Ifnar1*$^{-/-}$ mice infected with CCHFV Hoti (*Hawman et al., 2019*). Similar to the liver and plasma, in the kidneys and lungs, although early viral loads were similar between groups, by day 6 male mice infected with MA-CCHFV had higher viral loads compared to MA-CCHFV infected female mice or mice infected with CCHFV Hoti (*Figure 3D,E*). Lastly, both male and female mice infected with MA-CCHFV had significantly increased viral loads in the brain at days 3 through 8 PI, and male mice continued to have significantly increased viral RNA loads in the tissue to at least day 14 PI (*Figure 3F*). Cumulatively, these data indicate that the more severe disease seen in MA-CCHFV infected mice correlates with higher viral RNA burdens in multiple tissues.

## MA-CCHFV causes significant pathology in the livers of WT mice

CCHFV infection of humans typically results in a hemorrhagic-type disease with severe involvement of the liver. Histological examination of formalin-fixed sections of liver revealed that MA-CCHFV infection resulted in hepatocellular necrosis with acute inflammation in both male and female mice infected with MA-CCHFV by day 3 PI (*Figure 4A,B* and *Supplementary file 1*). However, consistent with prolonged clinical disease and delayed clearance of viral loads in the liver of male mice, male mice infected with MA-CCHFV had greater necrosis at day 6 and day 8 PI than infected female mice (*Figure 4B* and *Supplementary file 1*). Subacute hepatitis was also evident in MA-CCHFV-infected mice (*Figure 4B* and *Supplementary file 1*). Immunohistochemistry to detect viral antigen in the liver identified CCHFV antigen in liver endothelial cells, Kupffer cells, and hepatocytes in both male and female mice infected with MA-CCHFV at day 1 and day 3 PI (*Figure 4B* and *Supplementary file 1*). At day 6 PI, consistent with greater viral loads in male mice infected with MA-CCHFV, male mice had greater amounts of viral antigen present in the liver (*Figure 4B* and *Supplementary file 1*). By day 14 PI, all mice had cleared viral antigen from their livers. (*Figure 4B* and *Supplementary file 1*). Consistent with little-to-no clinical disease in CCHFV Hoti infected mice, in CCHFV Hoti infected mice little pathology was evident in the liver and viral antigen was cleared from the liver earlier than MA-CCHFV infected mice (*Supplementary file 1*). The complete histological and immunohistochemistry findings are provided in *Supplementary file 1*.

In addition to histological examination, we also evaluated liver enzymes in the blood. Compared to mock-infected mice, we observed a significant increase in liver enzymes in male mice infected with MA-CCHFV on days 3 and 6 PI (*Figure 4C,D*), consistent with the severe liver pathology in

**A**

Mock

**B**

**Male**
H&E          IHC

**Female**
H&E          IHC

1 DPI

3 DPI

6 DPI

4 DPI

**C**     **ALT**

**D**     **AST**

**Figure 4.** MA-CCHFV causes severe pathology in the livers of WT mice. (A–D) Groups of 8-week-old wild-type (WT) mice were infected were infected with 10,000 TCID$_{50}$ of MA-CCHFV or Hoti via the intraperitoneal (IP) route or mock-infected. (A) Representative liver sections from a mock-infected mouse is shown. (B) At indicated timepoints, MA-CCHFV-infected mice were euthanized, liver tissue fixed in formalin and paraffin embedded sections stained with H and E or an antibody against the CCHFV NP to identify viral antigen (IHC). Four mock-infected, four male and four female MA-CCHFV

*Figure 4 continued on next page*

Figure 4 continued
mice were evaluated at each timepoint and representative images shown. Images shown at ×200 magnification and scale bar indicates 100 μm. Study performed once. (C and D) At indicated timepoints, liver enzymes were measured in lithium heparin treated whole blood. ALT = Alanine aminotransferase, AST = Aspartate aminotransferase. N = 6 mock male and female, 4 Hoti-infected and 8 MA-CCHFV infected per group. Study performed once for Hoti and twice for mock and MA-CCHFV-infected mice. Data shown as mean plus standard deviation. Dashed line indicates upper limit of detection. Statistical comparison performed with two-way ANOVA with Tukey's multiple comparison test. p-Values between MA-CCHFV and respective sex Hoti-infected mice indicated with * for females, # for males and between MA-CCHFV male and MA-CCHFV female mice with +. *p<0.05, **p<0.01, ***p<0.001, ****p<0.0001.
The online version of this article includes the following figure supplement(s) for figure 4:

**Figure supplement 1.** Histology and IHC of spleens from MA-CCHFV infected mice.

these mice. Compared to mock-infected mice, female mice infected with MA-CCHFV had elevated liver enzymes at day 1 and day 3 PI (*Figure 4C,D*) but these levels were significantly less than those measured in MA-CCHFV infected male mice (*Figure 4C,D*). In agreement with the lack of overt clinical disease and little-to-no histological evidence of disease in the livers of Hoti-infected mice, no significant increases in liver enzymes were seen following infection of WT mice with CCHFV Hoti (*Figure 4C,D*). The complete blood chemistry data is provided in *Supplementary file 2*. Together this data demonstrates that similar to human CCHF cases, MA-CCHFV causes significant liver pathology in WT mice.

We also examined pathology in the spleen, kidney, lungs, and brains. Despite detectable viral RNA in these tissues, no lesions attributable to CCHFV were evident in the kidney, lung, and brain (*Supplementary file 1*). In the spleen, follicular, and red pulp necrosis was evident in both male and female MA-CCHFV-infected mice at day 6 PI (*Figure 4—figure supplement 1* and *Supplementary file 1*). Viral antigen was located primarily in the white and red pulp within mononuclear cells morphologically consistent with macrophages (*Figure 4—figure supplement 1*). These data demonstrate that in addition to the liver, pathology is also evident in spleens of MA-CCHFV infected mice.

## MA-CCHFV causes an inflammatory immune response

CCHFV infection of humans, NHPs and *Ifnar1*[-/-] mice results in production of inflammatory cytokines (*Zivcec et al., 2013*; *Bente et al., 2010*; *Hawman et al., 2019*; *Papa et al., 2006*; *Papa et al., 2016*; *Haddock et al., 2018a*). We therefore evaluated the plasma cytokine response in MA-CCHFV-infected WT mice. MA-CCHFV infection of WT mice resulted in production of multiple proinflammatory cytokines during acute disease including interleukin 1 beta (IL-1β), IL-5, IL-6, granulocyte colony-stimulating factor (G-CSF), KC (CXCL1), monocyte chemoattractant protein 1 (MCP-1, CCL2), macrophage inflammatory protein 1 alpha (MIP1α, CCL3), MIP1β (CCL4) and regulated on activation, normal T cell expressed and secreted (RANTES, CCL5) (*Figure 5*). Furthermore, more severe disease in male mice infected with MA-CCHFV was associated with significantly greater levels of IL-1β, IL-6, G-CSF, MCP-1, MIP1α, MIP1β, and RANTES compared to female infected mice (*Figure 5*). Notably, MCP-1 (CCL2) was rapidly upregulated in MA-CCHFV infected mice, with greater than 4500 pg/mL in the plasma of male and female mice by day 1 and strikingly, male mice had over 28,000 pg/mL in the plasma at day 3 PI (*Figure 5*). In agreement with little-to-no disease in Hoti-infected WT mice, these mice showed mostly transient increases in the cytokines evaluated (*Figure 5—figure supplement 1*). The complete profile of cytokines quantified by the 23-plex assay of mock, Hoti and MA-CCHFV infected mice is provided in *Figure 5—figure supplement 1*. Together these data demonstrate infection of WT mice with MA-CCHFV results in an inflammatory immune response.

## Type I IFN is required for survival in MA-CCHFV infected mice

The ability of clinical isolates of CCHFV to cause disease in type I IFN-deficient but not IFN-sufficient mice suggests non-adapted strains of CCHFV are unable to overcome mouse type I IFN responses. We hypothesized that MA-CCHFV may be able to replicate and cause disease in WT mice by avoiding or antagonizing production of type I IFN in vivo. We evaluated plasma IFNα and IFNβ levels in adult mock-infected or mice infected with either CCHFV strain Hoti or MA-CCHFV. Compared to



**Figure 5.** MA-CCHFV infection results in inflammatory cytokine responses in wild-type (WT) mice. Eight-week-old male or female WT mice were infected with 10,000 TCID$_{50}$ MA-CCHFV via the intraperitoneal (IP) route or mock-infected. At indicated timepoints, cytokine levels in the plasma was measured by 23-plex cytokine assay. Data shown as mean plus standard error. N = 6 mock male and female mice and seven to eight MA-CCHFV mice per sex per timepoint. Study performed twice. Statistical comparison performed with two-way ANOVA with Tukey's multiple comparison test. p-Values between MA-CCHFV and mock-infected mice indicated with * for females, # for males and between MA-CCHFV-infected male and female mice with +. *p<0.05, **p<0.01, ***p<0.001, ****p<0.0001.

The online version of this article includes the following source data and figure supplement(s) for figure 5:

**Source data 1.** Source data for Figure 5.

**Figure supplement 1.** Complete cytokine profile of mock, Hoti, or MA-CCHFV-infected mice.

mock-infected mice, we found that infection of WT mice with CCHFV resulted in significantly increased plasma levels of IFNα by day 1 PI (*Figure 6A*) and that levels among male or female mice infected with either MA or non-adapted CCHFV were similar (p>0.05) (*Figure 6A*). Male mice infected with MA-CCHFV still had significantly elevated levels of IFNα at day 3 PI (*Figure 6A*). In contrast, all other groups had no significant IFNα response above our limit of detection (250 pg/mL) at day 3 PI or later (*Figure 6A*). Interestingly, male but not female mice infected with MA-CCHFV had significant amounts of IFNβ at day 3 PI (*Figure 6B*) suggesting MA-CCHFV infection in male mice elicits production of IFNα followed by IFNβ. Female mice infected with MA-CCHFV or mice infected with CCHFV Hoti had no significant IFNβ response at any timepoint evaluated following infection (*Figure 6B*).

Since CCHFV induced a rapid type I IFN response in WT mice and we have previously demonstrated that *Ifnar1*[-/-] mice infected with CCHFV Hoti succumb to the infection (*Hawman et al., 2018*), we sought to determine if type I IFN was similarly required to survive MA-CCHFV infection. We infected *Ifnar*[-/-] mice with MA-CCHFV and found that male and female *Ifnar1*[-/-] mice infected with MA-CCHFV rapidly lost weight and succumbed to the infection with a mean-time-to-death



**Figure 6.** Type I IFN is required for survival following MA-CCHFV infection. (A–B) Eight-week-old male or female wild-type (WT) mice were infected with 10,000 $TCID_{50}$ of CCHFV Hoti, MA-CCHFV via the intraperitoneal (IP) route or mock-infected. At indicated timepoints, plasma IFNα (all subtypes) (A) or IFNβ (B) was quantified by ELISA. N = 4–8 per group. Study performed once for Hoti and twice for mock and MA-CCHFV. Data shown as mean plus standard deviation. Dashed line indicates limit of detection determined from manufacturer provided standard curve. Statistical comparison performed using two-way ANOVA with Tukey's multiple comparison test. (C–D) Groups of 8-week-old male or female WT mice or 10- to 13-week-old *Ifnar1*[-/-] mice were infected with 10,000 $TCID_{50}$ of MA-CCHFV via the IP route. Mice were weighed daily (C) and monitored for survival (D). N = 4–5 per group. Study performed once. Data shown as mean plus standard deviation. Statistical comparison between

*Figure 6 continued on next page*

*Figure 6 continued*

*Ifnar1*[-/-] and respective sex WT mice performed using two-way ANOVA with Sidak's multiple comparison test (**C**) or Log-rank test with Bonferroni's correction (**D**). *p<0.05, **p<0.01, ****p<0.0001.

The online version of this article includes the following source data for figure 6:

**Source data 1.** Source data for Figure 6.

(MTD) of day 6 and day 3 PI, respectively (*Figure 6C,D*). Cumulatively, these data demonstrate that MA-CCHFV infection induces similar early type I IFN responses as parental CCHFV strain Hoti in vivo and that type I IFN is necessary to survive acute MA-CCHFV infection.

## MA-CCHFV causes lethal disease in mice deficient in adaptive immunity

WT male mice infected with MA-CCHFV began recovering around day 7 PI (*Figure 1A*), about when early adaptive immune responses might be engaged against CCHFV (*Hawman et al., 2019*). When we evaluated CCHFV-specific antibody responses by ELISA, both male and female mice infected with MA-CCHFV developed significant IgM and IgG responses against CCHFV by day 6 PI (*Figure 7A,B*). We also evaluated T-cell responses against the CCHFV nucleoprotein (NP) by IFNγ ELISpot. By day 14 PI, both male and female mice infected with MA-CCHFV had significant T-cell responses against NP (*Figure 7C*). These data demonstrate that MA-CCHFV infection elicits both humoral and cellular immune responses against CCHFV.

To determine the contribution of these responses to recovery from MA-CCHFV infection, we infected 8-week-old WT or B- and T-cell-deficient *Rag1*[-/-] mice with MA-CCHFV. Male WT or *Rag1*[-/-] mice infected with MA-CCHFV showed similar weight loss through day 5 PI but *Rag1*[-/-] mice exhibited significantly greater weight loss at day 6 PI and later (*Figure 7D*). Whereas male WT mice began to recover on day 7, infected *Rag1*[-/-] mice continued to decline and all succumbed to the infection with an MTD of day 9 PI (*Figure 7E*). Interestingly, the one male *Rag1*[-/-] mouse that succumbed on day 10 was found to have ataxia and hindlimb weakness just prior to euthanasia suggesting possible neurological involvement. Female *Rag1*[-/-] mice infected with MA-CCHFV began exhibiting significantly greater weight loss than WT mice by day 6 PI (*Figure 7D*) and all succumbed to the infection with an MTD of day 7 PI (*Figure 7E*). These data indicate that both male and female mice require adaptive immunity to survive MA-CCHFV infection.

## Sequencing of MA-CCHFV

We sequenced our stock of parental CCHFV Hoti, MA-CCHFV and intermediate variants at passages 4 and 9 to determine what mutations had accumulated in the viral genome during passaging in mice (*Table 1*). CCHFV is a negative-sense RNA virus with three genomic segments, small (S), medium (M), and large (L). Sequencing identified mutations in all three viral segments of MA-CCHFV (*Table 1*). S encodes for NP and a small non-structural protein (NSs) in an opposite sense open reading frame (*Zivcec et al., 2016*). Two mutations in the S segment were identified: mutation at nucleotide (nt) A739G (Hoti > MA-CCHFV) resulting in amino acid (aa) change NP I228M and nt A806G resulting in aa NP K251E (*Table 1*). Mutation nt A806G also results in an aa F26S coding change of the CCHFV NSs. Conversely, the nt A709G mutation does not result in a coding change in the NSs protein (*Table 1*).

The M segment encodes the glycoprotein precursor (GPC) that is proteolytically processed to produce a heavily glycosylated mucin-like domain (MLD), GP38 accessory protein, the envelope glycoproteins Gn and Gc and the medium non-structural protein NSm (*Zivcec et al., 2016*). Three mutations were identified in the CCHFV M segment. Amino acid change C865Y was identified in the NSm protein (*Table 1*). In addition, two synonymous nucleotide changes were identified, nt A1502G and nt T4068C, located in the GP38 accessory protein and Gc glycoprotein, respectively.

The L segment of CCHFV at 12 kb long is uniquely large among members of the *Bunyavirales* order, encoding a protein of over 3900 amino acids. It encodes the viral RNA-dependent RNA-polymerase (RdRp), along with domains for a zinc finger and leucine zipper (*Honig et al., 2004*). At the 5' end, it encodes an ovarian-tumor like (OTU) domain that has been shown to have de-ISGylation and de-ubiquitination function that are critical for viral replication (*Scholte et al., 2017*; *Zivcec et al., 2016*). However, given the large size of the CCHFV L protein, it likely possesses

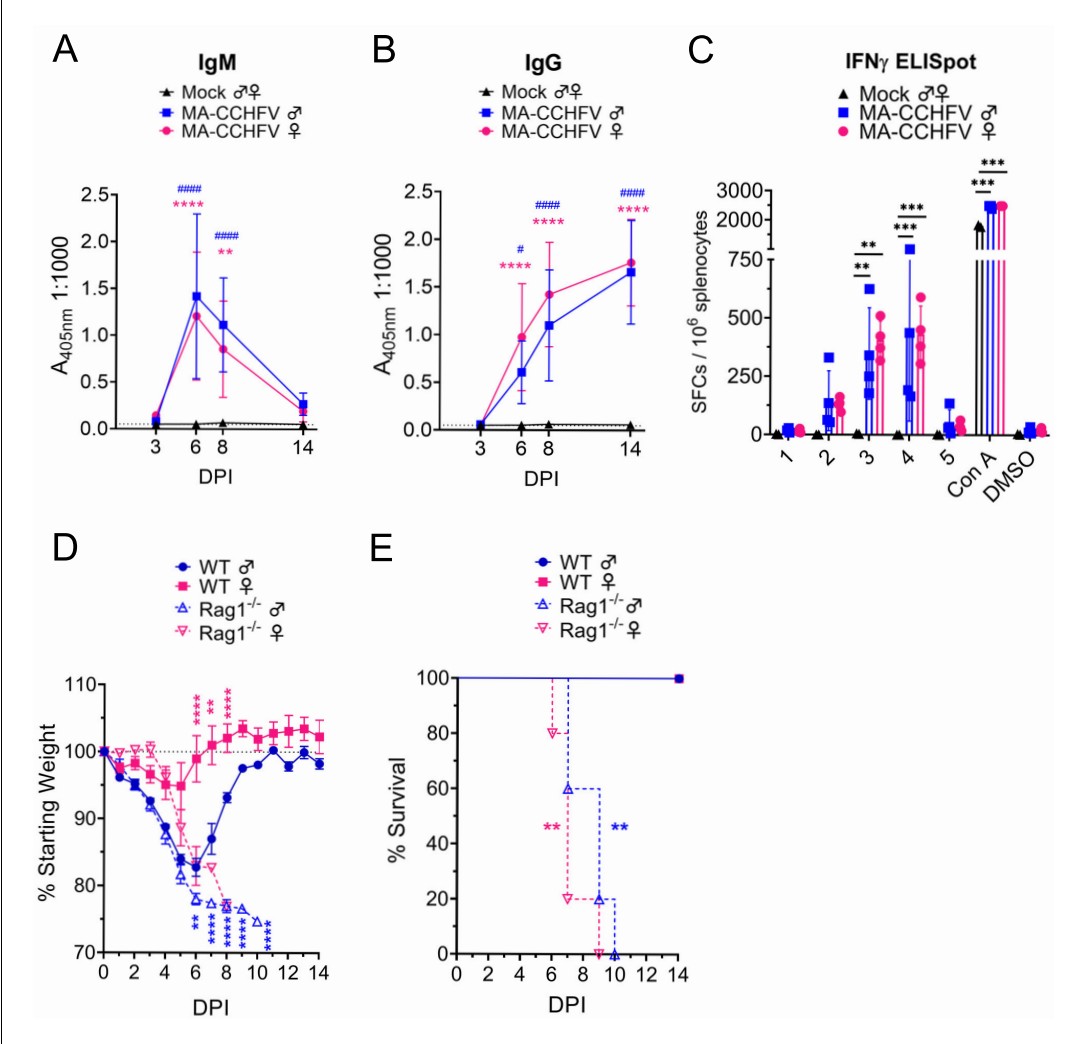

**Figure 7.** MA-CCHFV is lethal in mice lacking adaptive immunity. (A–C) Groups of 8-week-old wild-type (WT) mice were infected were infected with 10,000 TCID$_{50}$ of MA-CCHFV via the intraperitoneal (IP) route or mock-infected. At indicated timepoints CCHFV-specific IgM (A) or IgG (B) in the plasma was measured by whole-virion ELISA. N = 5–6 per timepoint for mock and 7–8 per timepoint for MA-CCHFV. Study performed twice. Data shown as mean plus standard deviation. (C) At day 14 PI, T-cell responses in the spleen were measured by IFNγ ELISpot. Splenocytes were stimulated with overlapping peptide pools derived from the CCHFV NP (1 – 5), concanavalin A (Con A) or DMSO-vehicle alone. N = 2 for mock and four for MA-CCHFV infected. Study performed once. Data shown as mean plus standard deviation. (D and E) Groups of 8-week-old WT or *Rag1⁻/⁻* mice were infected with 10,000 TCID$_{50}$ of MA-CCHFV via the IP route. Mice were weighed daily (C) and monitored for survival (D). N = 5 per group. Study performed once. Data shown as mean plus standard deviation. Statistical comparison between *Rag1⁻/⁻* and respective sex WT mice performed using two-way ANOVA with Sidak's multiple comparison test (C) or Log-rank test with Bonferroni's correction (D).

The online version of this article includes the following source data for figure 7:

**Source data 1.** Source data for Figure 7.

additional functions in the viral life cycle. Two non-synonymous (nt G6097A, aa S2007N and nt C9919T, aa P3281L) and two synonymous (nt C8135T, aa V2686V and nt G11618A, aa E3847E) mutations were identified in the viral L segment (*Table 1*). With the exception of the synonymous mutation nt C8135T, the mutations in the L segment are in regions of the L protein without precisely described function. Nt C8135T resulting in a synonymous V2686V is located within the catalytic RdRp domain of the L segment (*Zivcec et al., 2016*; *Honig et al., 2004*), although it is unlikely the synonymous coding change has functional consequence toward this activity.

**Table 1.** Mutations identified in MA-CCHFV.

| Segment | SNP (Hoti>Mutant) | Coding Change (Hoti>Mutant) | Domain | Mutant Frequency (%) Passage 4 | Passage 9 | MA-CCHFV | % AA Conservation among CCHFV Strains |
|---|---|---|---|---|---|---|---|
| S | A739G | NP I228M | Arm | 93 | 87 | 88 | 100 (I) |
| | A806G | NP K251E and NSs F26S | Arm and Unknown | 95 | 100 | 99 | NP 100 (K); NSs 88 (L), 12 (F) |
| M | A1502G | R475R | GP38 | 3 | 35 | 54 | |
| | G2671A | C865Y | NSm | 100 | 100 | 100 | 100 (C) |
| | T4068C | L1331L | Gc | 98 | 100 | 100 | |
| L | G6097A | S2007N | Unknown | 83 | 97 | 98 | 71 (S), 29 (N) |
| | C8135T | V2686V | RdRp | 83 | 95 | 96 | |
| | C9919T | P3281L | Unknown | 85 | 96 | 96 | 100 (P) |
| | G11618A | E3847E | Unknown | 85 | 98 | 97 | |

Interestingly, with the exception of the nt A1502G mutation in the M segment, all mutations present in the MA-CCHFV stock (passage 11) were also present in the passage 4 stock at a frequency of >80% of reads (*Table 1*). These data suggest that passaging of CCHFV in *Rag2*$^{-/-}$ mice quickly selected for mouse adapted variants. We also evaluated the conservation of the mutated aa residues in MA-CCHFV among seven divergent CCHFV isolates from all five clades of CCHFV (*Lukashev et al., 2016*). We found that the mutated residues in MA-CCHFV NP, NSm and L occurred at highly conserved residues among divergent CCHFV strains (*Table 1*). The mutation F26S in MA-CCHFV NSs occurred at a unique F26 residue in parental strain Hoti as all other CCHFV strains evaluated possessed a leucine at this residue (L26) (*Table 1*). Interestingly, the L protein S2007N mutation in MA-CCHFV mutated the Hoti S2007 residue to an N, a residue also possessed by the L proteins of CCHFV strains Oman and UG3010 (*Table 1*).

## Discussion

In this report, we have described a novel mouse-adapted variant of CCHFV capable of causing a severe inflammatory disease in adult, immunocompetent mice. To our knowledge, this represents the first CCHFV variant capable of causing overt disease in wild-type mice. To date, infection of adult immunocompetent mice with CCHFV has resulted in severely restricted viral replication and little to no disease (*Zivcec et al., 2013*; *Bente et al., 2010*). Even strains passaged 27 times in newborn mice failed to cause disease in adult immunocompetent mice (*Hoogstraal, 1979*) and as a result, studies of severe CCHF have required use of mice either genetically deficient in type I IFN signaling (e.g. *Ifnar1*$^{-/-}$) (*Zivcec et al., 2013*; *Bente et al., 2010*; *Hawman et al., 2018*; *Hawman et al., 2019*; *Oestereich et al., 2014*) or transiently suppressed by type I IFN receptor blockade (*Garrison et al., 2017*; *Lindquist et al., 2018*). Thus, until now, the only immunocompetent animal model of CCHF has been cynomolgus macaques (*Haddock et al., 2018a*) and ethical and practical considerations limit the use of this model for initial investigations of CCHFV pathogenesis or therapeutics. The ability of MA-CCHFV to cause disease in fully immunocompetent mice represents a significant improvement in our ability to study CCHFV pathogenesis in a highly tractable mouse model. Importantly, MA-CCHFV recapitulates many aspects of severe human CCHF, with MA-CCHFV-infected mice developing high viral loads in multiple tissues, severe liver pathology and an inflammatory immune response consistent with human cases of CCHF. More severe disease in male mice was associated with higher viral loads, increased liver enzymes and increased inflammatory cytokines compared to female mice or mice infected with non-adapted CCHFV Hoti. These parameters have all been shown to correlate with disease outcome in humans (*Bente et al., 2013*; *Ergönül, 2006*; *Ergonul et al., 2006*; *Papa et al., 2006*; *Papa et al., 2016*) suggesting the spectrum of disease severity in WT mice has similar correlates to human CCHF cases.

The consistent sex-linked bias toward more severe disease in adult WT male mice was unexpected. Our data show that adult female WT mice are largely resistant to severe disease following infection with MA-CCHFV, exhibiting milder clinical disease, earlier control of viral loads, reduced

inflammatory cytokine production and reduced liver pathology compared to male mice. However, MA-CCHFV infection was lethal in young-female WT mice and female *Ifnar1*[-/-] and *Rag1*[-/-] mice demonstrating that resistance to MA-CCHFV by female mice is age-dependent and requires both innate and adaptive host responses. Nevertheless, the relevance of this sex-linked bias in mice toward human disease is unclear. Although some studies have identified human males as more likely to become infected with CCHFV (*Monsalve-Arteaga et al., 2020*; *Yagci-Caglayik et al., 2014*; *Bower et al., 2019*; *Chinikar et al., 2010*), this is more likely due to cultural practices in which men are more likely to engage in activities such as farming, herding or butchering that place them at higher risk for exposure to CCHFV (*Chapman et al., 1991*; *Gunes et al., 2009*; *Ozkurt et al., 2006*).

Alternatively, female mice and humans often exhibit stronger immune responses to vaccinations and pathogens (*Klein and Flanagan, 2016*) and it is possible that female mice exhibit more robust and/or protective responses to the MA-CCHFV infection. Despite similar viral loads between male or female mice infected with MA-CCHFV at day 1 and day 3 PI, male mice had significantly greater production of inflammatory cytokines IL-6, G-CSF, MCP-1, MIP1α, MIP1β, RANTES, and IFNβ than female mice on day 3 PI demonstrating that male mice responded with a much stronger inflammatory response than female mice. This response in male mice was associated with delayed viral control and more severe disease suggesting these responses may contribute to the distinct disease outcome observed between male and female mice. Similar cytokine patterns have been observed in Ebola virus disease where dysregulated host responses significantly contribute to the severe mortality observed in animal models and humans (*Bixler and Goff, 2015*). Furthermore, IFNβ has been shown to have immunosuppressive effects leading to impaired viral clearance (*Ng et al., 2015*) suggesting the distinct type I IFN response in MA-CCHFV infected male mice may also have consequence during later stages of disease. In contrast to innate responses, male and female mice developed similar B- and T-cell responses to the infection and our data indicate these responses are critical for survival of acute MA-CCHFV infection. Further studies will be needed to determine how these responses limit and/or promote MA-CCHFV pathogenesis. Nevertheless, our findings clearly highlight how MA-CCHFV infection of WT mice provides a suitable model for studying innate and adaptive immune responses to CCHFV infection, including type I IFN responses, studies that have been severely limited with current mouse models of CCHF.

We also observed differences in disease outcome upon infection of several strains of mice, with 129S1 mice appearing largely resistant to MA-CCHFV. Thus, in addition to sex-linked determinants, there also exist strain-linked determinants of disease outcome. Given the wide-spectrum of disease outcomes in humans and NHPs infected with CCHFV (*Ergönül, 2006*; *Hawman et al., 2020*; *Haddock et al., 2018a*), the spectrum of disease in male vs female WT mice and between different laboratory strains of mice provides a novel opportunity to investigate the host determinants of CCHF disease outcome in a mouse model.

Cumulatively, we identified five non-synonymous mutations in MA-CCHFV compared to parental CCHFV Hoti strain. The function of the mutations identified in MA-CCHFV will require further study. Two mutations were identified in the viral S segment resulting in two coding changes in NP and one also resulting in a coding change in the opposite sense encoded NSs. The CCHFV NSs has been shown to disrupt the mitochondrial membrane potential and induce apoptosis (*Barnwal et al., 2016*). Given the central role of mitochondria in innate immune signaling (*West et al., 2011*) it is tempting to speculate that the coding mutation identified in the MA-CCHFV NSs may modulate the ability of MA-CCHFV to antagonize mouse innate immune signaling. Indeed, the NSs of several distantly related viruses in the *Bunyavirales* order have been shown to block the host type I IFN response (*Billecocq et al., 2004*; *Wuerth and Weber, 2016*; *Ly and Ikegami, 2016*; *Rezelj et al., 2017*) demonstrating this may be a common function of *Bunyavirales* NSs proteins. However, to date, such a function of the CCHFV NSs has not been described. The NP of CCHFV is responsible for binding the viral RNA but also has functions in promoting translation (*Jeeva et al., 2017*), has an endonuclease function (*Guo et al., 2012*), interacts with human MxA, a potent restriction factor of CCHFV in vitro (*Andersson et al., 2004*) and contains a conserved DEVD caspase 3 cleavage site (*Carter et al., 2012*; *Karlberg et al., 2011*). Structurally, the mutations in NP are in the mobile 'arm' domain of NP, proximal (37 and 14aa distant) to the DEVD caspase 3 cleavage site (*Carter et al., 2012*; *Guo et al., 2012*). Thus, the mutations in the MA-CCHFV NP could alter several functions of the NP protein.

One non-synonymous amino acid change, C865Y, was identified within the NSm protein of the CCHFV M segment. The precise function of the CCHFV NSm in the viral life cycle is unknown but the selection for and complete penetrance of the C865Y mutation in the MA-CCHFV NSm suggests CCHFV NSm has critical function in vivo for the MA phenotype. In distantly related Bunyamwera virus, the NSm protein is required for virus assembly (*Shi et al., 2006*), although in Rift Valley Fever Virus, also distantly related to CCHFV, it is dispensable for virus replication (*Gerrard et al., 2007*) indicating members of the *Bunyavirales* order encode NSm proteins of varied function. Recently, CCHFV lacking the NSm protein was found to grow to similar titers in vitro and in vivo in IFNAR$^{-/-}$ mice demonstrating NSm is not required for viral growth in the absence of type I IFN (*Welch et al., 2020*). In agreement, in vitro virus-like-particle studies showed NSm supported efficient virion assembly and secretion although NSm was not essential for these functions (*Freitas et al., 2020*). In addition, NSm may also play a role in the tick-vector as growth of a mouse-passaged strain of CCHFV in ticks selected for a mutation in NSm demonstrating tick-specific selective pressures are exerted on NSm (*Xia et al., 2016*). Notably, we did not identify any coding change in the viral Gn or Gc envelope glycoproteins suggesting MA-CCHFV has not adapted to mice by altering utilization of mouse-specific proteins for binding and entry.

Four mutations resulting in two non-synonymous mutations in the L protein were found in MA-CCHFV L segment. The L segment of CCHFV is unusually large for viruses of the *Bunyavirales* order (*Zivcec et al., 2016*) encoding a protein of nearly 4000 amino acids and the two coding mutations found in MA-CCHFV occur in regions without ascribed function. Thus, it is difficult to speculate on the functional consequence of the mutations identified in the L segment and further studies will be needed. Interestingly, despite the function of the L protein OTU domain in modulating innate immunity (*Scholte et al., 2017*; *Capodagli et al., 2013*; *Deaton et al., 2016*; *Frias-Staheli et al., 2007*; *James et al., 2011*) and the hypothesis that poor affinity of the CCHFV OTU domain for mouse ISG-15 could be a barrier to CCHFV infection in mice (*Deaton et al., 2016*), no mutations were identified in or proximal to this domain.

The availability of a reverse genetics system for CCHFV (*Bergeron et al., 2015*) will allow for further investigation into the function of these mutations in the MA-CCHFV phenotype. Additionally, although our study was focused on developing a tractable small rodent model for CCHF, MA-CCHFV furthers the suggestion that distinct hosts and reservoirs can influence CCHFV genetics and potential virulence (*Xia et al., 2016*; *Gonzalez et al., 1995*). The availability of tools to study CCHFV transmission *in vivo* under requisite biocontainment (*Gargili et al., 2013*) may enable studies using MA-CCHFV to explore the evolutionary constraints placed on CCHFV by its tick-mammal life cycle.

In conclusion, MA-CCHFV represents a significant advancement for research into CCHFV by enabling study of CCHFV infection in adult, immunocompetent mice while importantly still recapitulating many aspects of severe cases of human CCHF. The ability to infect the plethora of genetically manipulated mouse strains available will permit studies investigating pathogenic and protective host responses to the infection, including those requiring intact type I IFN signaling. These studies may identify novel therapeutic intervention strategies to limit the severe morbidity and mortality observed in CCHFV-infected humans. MA-CCHFV will also enable preclinical evaluation of vaccines and antivirals in mice that are fully competent for type I interferon, an improvement over existing models requiring genetic- or transient-IFN deficiency. Lastly, ongoing studies evaluating the function of the identified mouse-adaptive mutations in mediating the mouse-adapted phenotype will further our understanding of the functions of viral proteins in antagonism of host immune responses.

# Materials and methods

## Key resources table

| Reagent type (species) or resource | Designation | Source or reference | Identifiers | Additional information |
|---|---|---|---|---|
| Other | Hoti | This paper | | Virus strain |
| Other | MA-CCHFV | This paper | | Virus strain |

## Mice

*Rag2*<sup>-/-</sup> and *Rag1*<sup>-/-</sup> mice on the C57BL/6J background (stock #008449, stock #002216 respectively), wild-type C57BL/6J (stock #000664), BALBc/J (stock #000651), and 129S1 (stock #002448) were purchased from Jackson Laboratories. C57BL6/NCr (strain code 027) and outbred CD1 mice (strain code 022) were purchased from Charles River Laboratories. *Ifnar1*<sup>-/-</sup> mice on the C57BL/6J background were from an in-house breeding colony. Mice were randomly assigned to study groups. Unless otherwise indicated, mice were all 6–8 weeks of age at time of infection except for the first passage in wild-type mice (passage 10) which utilized mice 3 weeks of age. Mice were humanely euthanized according to the following criteria: ataxia, extreme lethargy (animal is unresponsive to touch), bloody discharge from nose, mouth, rectum or urogenital area, tachypnea, dyspnea or paralysis of the limbs. Although animals were comprehensively evaluated for the above signs, animals that succumbed following MA-CCHFV infection were typically euthanized for extreme lethargy and dyspnea and in one mouse, ataxia was also present. For survival analysis, mice euthanized for severe disease were recorded as having succumbed +1 day to day of euthanasia. For $ID_{50}$ calculations, mice with detectable anti-CCHFV Ig ($A_{405nm}$ >0.2) at a 1:400 dilution of serum were considered productively infected and $ID_{50}$ calculated using the *Reed and Muench, 1938* method.

## Tissue passaging

Mice were humanely euthanized, and piece of liver collected into a tube. A steel bead and 1 mL of L-15 media (ATCC) supplemented with 10% fetal bovine serum (FBS) and penicillin/streptomycin added. Tissue was homogenized at 30 hz for 1 min in a TissueLyser (Qiagen) then briefly spun at maximum RPM in a benchtop centrifuge to pellet large debris. Clarified tissue homogenate was then innoculated into naive mice via the IP route.

## Virus stocks and deep sequencing

Parental strain CCHFV Hoti was grown, titered, and sequenced as previously described (*Hawman et al., 2018*; *Haddock et al., 2018a*). MA-CCHFV or intermediate variants were grown by inoculation of clarified liver tissue homogenate onto an SW13 cell monolayer and supernatant harvested 48 hr later. Stocks were generated on SW13 cells purchased from the ATCC and used at passage 11. Cell identity was not authenticated. Supernatant was clarified, aliquoted, and titered by SW13 median tissue culture infectious dose ($TCID_{50}$) assay. Virus stocks were sequenced with Illumina MiSeq-based deep sequencing to exclude contamination, including mycoplasma or other viral pathogens, and to identify mutations present. The sequence of MA-CCHFV has been deposited to Genbank (Accession #s MW058028 – MW058030). Mutations in mouse passaged CCHFV were identified by comparison to parental strain CCHFV Hoti sequenced in parallel. Minimum read depth at any mutation identified in the MA-CCHFV stock was 152 reads. For sequence comparison, we compared MA-CCHFV sequence to Afghan09 (Genbank Accession #s: HM452305, HM452306, HM452307), ArD15786 (DQ211614, DQ211627, DQ211640), IbAr10200 (MH483987, MH483988, MH483989), Turkey 2004 (KY362515, KY362517, KY325619), Oman (KY362514, KY362516, KY362518), UG3010 (DQ211650, DQ211637, DQ211624), and Hoti (MH483984, MH483985, MH483986).

## Blood chemistry

At time of euthanasia, whole blood was collected into lithium heparin treated tubes and blood chemistry analyzed with Preventive Care Profile Plus disks on Vetscan two analyzers (Abaxis). The complete blood chemistry data is available in the supplemental materials.

## Cytokine analysis

At time of euthanasia, whole blood was collected into lithium heparin treated tubes (BD) via cardiac puncture. Plasma was separated by centrifugation and irradiated according to approved procedures to inactivate CCHFV. Plasma cytokine levels were analyzed by 23-plex mouse cytokine assay according to manufacturer's instructions (Biorad). Plasma IFNα (all sub-types) and IFNβ levels were quantified in 1:10 dilutions of plasma by ELISA according to manufacturer's instructions (PBL Assay).

## qRT-PCR

RNA from mouse plasma was isolated using Qiamp RNA-mini isolation kit (Qiagen) and RNA from tissues isolated using the RNeasy mini isolation kit (Qiagen). Viral loads were quantified by qRT-PCR as follows: primers and probe specific for the CCHFV S segment: Forward: 5'- TCTACATGCACCC TGCTGTG, Reverse: 5'- AGCGTCATCAGGATTGGCAA and probe 5'- TGGGTGTCTGC TTTGGAACA were used in a one-step qRT-PCR reaction with either Quantifast reagents (Qiagen) for tissue RNA or LightCycler 480 RNA Master Hydrolysis Probes (Roche) for plasma RNA samples. Probe was labeled with a 5' 6-FAM, ZEN internal quencher and 3' Iowa Black quencher. Primers and probes were purchased from Integrated DNA Technologies. Reactions were run on a Quantstudio 3 or 5 instrument (ThermoFisher). Cycling conditions for Quantifast reagents were: 50°C for 10 min, 95°C for 5 min and 40 cycles of 95°C for 10 s, 60°C for 30 s. Cycling conditions for LightCycler 480 reagents were 61°C for 3 min, 95°C for 30 s and 45 cycles of 95°C for 10 s, 60°C for 30 s and 72°C for 1 s. An in vitro transcribed RNA standard curve was generated by T7 runoff transcripts of the CCHFV S segment and included in every run.

## ELISA

An ELISA to detect anti-CCHFV Ig was performed as previously described (*Hawman et al., 2019*) with an anti-mouse Ig detection antibody (Southern Biotech) to detect all immunoglobulin isotypes or anti-mouse IgG or IgM (Southern Biotech) to measure specific isotypes.

## IFNγ ELISpot

An IFNγ ELISpot on splenocytes stimulated with peptides derived from the CCHFV NP was performed as before (*Hawman et al., 2019*).

## Histology and IHC

Tissues were fixed in 10% neutral buffered formalin with two changes for a minimum of 7 days. Tissues were placed in cassettes and processed with a Sakura VIP-6. Tissue Tek on a 12 hr automated schedule, using a graded series of ethanol, xylene, and PureAffin (Cancer Diagnostics). Embedded tissues were sectioned at 5 µm and dried overnight at 42°C prior to staining. Specific anti-CCHF immunoreactivity was detected using a rabbit anti-CCHF NP (IBT Bioservices) at a 1:2000 dilution as the primary antibody and Vector Laboratories ImPRESS-VR anti-rabbit IgG polymer kit (catalog no. MP-6401) neat as the secondary antibody. The tissues were processed for immunohistochemistry using the Ventana Ultra automated stainer using the Roche Tissue Diagnostics Discovery Chromo-Map DAB detection kit (catalog no. 760–159). Tissue sections were scored by certified pathologists who were blinded to study groups.

## Acknowledgements

We thank the Rock Mountain Laboratories Veterinary Branch, Research Technology Branch Genomics and Histology core for their support of these studies. This study was supported by the Intramural Research Program of the NIAID/NIH and by the European Union's Horizon 2020 program (CCHVaccine consortium, agreement no. 732732). Funders had no role in study design, data interpretation or decision to publish.

## Additional information

### Funding

| Funder | Author |
| --- | --- |
| NIAID | David W Hawman |

The funders had no role in study design, data collection and interpretation, or the decision to submit the work for publication.

## Author contributions
David W Hawman, Conceptualization, Data curation, Formal analysis, Supervision, Validation, Investigation, Visualization, Methodology, Writing - original draft, Project administration, Writing - review and editing; Kimberly Meade-White, Investigation, Methodology; Shanna Leventhal, Friederike Feldmann, Investigation; Atsushi Okumura, Brian Smith, Dana Scott, Formal analysis, Investigation; Heinz Feldmann, Resources, Supervision, Funding acquisition, Investigation, Methodology, Project administration, Writing - review and editing

## Author ORCIDs
David W Hawman https://orcid.org/0000-0001-8233-8176

## Ethics
Animal experimentation: All procedures with infectious CCHFV were conducted at biosafety level 4 (BSL4) conditions in accordance with operating procedures approved by the Rocky Mountain Laboratories institutional biosafety committee. Animal experiments were approved by the institutional animal care and use committee, protocol #s 2017-68 and 2019-63. Studies performed by experienced personnel under veterinary oversight. Mice were group-housed in HEPA-filtered cage systems and acclimatized to BSL4 conditions prior to start of the experiment. They were provided with nesting material and food and water ad libitum.

## Decision letter and Author response
Decision letter https://doi.org/10.7554/eLife.63906.sa1
Author response https://doi.org/10.7554/eLife.63906.sa2

# Additional files

## Supplementary files
• Supplementary file 1. Complete histological and IHC findings of formalin-fixed tissue sections from CCHFV-infected mice.

• Supplementary file 2. Complete blood chemistry of CCHFV-infected mice.

• Transparent reporting form

## Data availability
Relevant source data for figures are provided and the consensus sequence of MA-CCHFV has been deposited to Genbank (Accession #s MW058028 - MW058030).

The following datasets were generated:

| Author(s) | Year | Dataset title | Dataset URL | Database and Identifier |
|---|---|---|---|---|
| Hawman DW, Meade-White K, Leventhal S, Feldmann F, Okumura A, Smith B, Scott D, Feldmann H | 2021 | Crimean-Congo hemorrhagic fever orthonairovirus strain MA/CCHFV segment L, complete sequence | https://www.ncbi.nlm.nih.gov/nuccore/MW058028 | NCBI GenBank, MW058028 |
| Hawman DW, Meade-White K, Leventhal S, Feldmann F, Okumura A, Smith B, Scott D, Feldmann H | 2021 | Crimean-Congo hemorrhagic fever orthonairovirus strain MA/CCHFV segment S, complete sequence | https://www.ncbi.nlm.nih.gov/nuccore/MW058030 | NCBI GenBank, MW058030 |
| Hawman DW, Meade-White K, Leventhal S, Feldmann F, | 2021 | Crimean-Congo hemorrhagic fever orthonairovirus strain MA/CCHFV segment M, complete sequence | https://www.ncbi.nlm.nih.gov/nuccore/MW058029 | NCBI GenBank, MW058029 |

Okumura A, Smith
B, Scott D,
Feldmann H

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
