## [Decision Letter]

**Acceptance summary:**

This study presents the first description of a lethal mouse model of CCHFV in immunocompetent mice. Development of a mouse-adapted strain of CCHFV, along with identification of differences in disease susceptibility based on sex of the animal, represent an important finding in the field of emerging viruses. The work is extremely well done, with defined study aims, clearly presented results and robust data covering the major aspects of disease pathology. Furthermore, the findings presented here are important given the interest and need for an immunocompetent small animal model of CCHFV disease. This work will greatly benefit the field in both basic understanding of CCHFV pathology as well as in ongoing antiviral and vaccine development.

**Decision letter after peer review:**

Thank you for submitting your article "Immunocompetent Mouse Model for Crimean-Congo Hemorrhagic Fever Virus" for consideration by *eLife*. Your article has been reviewed by three peer reviewers, one of whom is a member of our Board of Reviewing Editors, and the evaluation has been overseen by Miles Davenport as the Senior Editor. The following individual involved in review of your submission has agreed to reveal their identity: Dennis Bente (Reviewer #3).

The reviewers have discussed the reviews with one another and the Reviewing Editor has drafted this decision to help you prepare a revised submission.

Summary:

The development of a suitable animal model that mimic the human-specific disease Crimean Congo hemorrhagic fever has been difficult since the discovery of the disease in 1944. Immunocompetent small mammalian models are permissive to infection but refractory to disease. As a result, mice with knock outs in the interferon response have been used, which are not ideally suited of the immune system and vaccines. Hawman et al. present and characterize an immunocompetent mouse model for CCHF by adapting the CCHFV to the mouse. This is a valuable tool in pathogenesis, countermeasure, and virus evolution studies. This straightforward, well-written manuscript that also describes important sex differences in susceptibility to disease in mice infected with the mouse-adapted CCHFV. Overall, this study is a significant step forward for the field through its description of an immunocompetent mouse model. Comments to improve the text are listed below.

Revisions:

1) One critique of the study is the lack of data detailing the clinical disease of MA-CCHFV itself. While weight loss is a useful metric in this model, I would have liked to have seen an evaluation of clinical signs throughout the challenge period, i.e., clinical scores at each day, which would help the reader better understand any inherent variability in the model. For example, and it is touched upon in the results, the large error bars in Figure 1B indicate that the female mice infected with MA-EBOV have a much more variable disease course than male mice – a representation of clinical scores would help with the reporting of this data. It may also be useful to represent individual animals in these weight loss graphs rather than use the mean, again for clarity. A clearer understanding of the expected disease course and clinical signs in a MA-CCHFV infection from challenge to recovery is would benefit any future research understanding the efficacy of potential antiviral compounds or vaccines.

2) Why were the passaged viruses not plaque -picked, and can the authors speculate on how the viral swarm/quasi species is influenced by the passaging? The authors comment on the non-synonymous mutations but not really how the SNP change from passage to passage.

3) Results first paragraph: can the authors include the onset of disease for each passage in the supplementary information?

4) Was there a reason than an intermediate dose (1,000 TCID50s) of MA-CCHFV was used when infecting multiple mouse strains rather than the 10,000 TCID50s used elsewhere?

5) While MA-CCHFV was shown to be lethal in the 3wk old mice, do we have comparative data for for wt Hoti?

6) The authors state that the clinical presentation was milder in females. I might have overlooked this in the manuscript, but is there a statistical side-by-side comparison of males vs. females in their different clinical parameters (weight loss, disease severity/scoring)? Figure 1?

7) The authors state "NSm is dispensable for virus replication (Gerrard et al., 2007)". This is true for mammalian systems, however, not in mosquitoes, where it determines vector competence. Han et al., 2016 demonstrated a non-synonymous amino acid change in NSm after the mouse-passaged CCHFV strain IbAr 10200 was passaged back in ticks. This could be thought of a “reversion” back to a virus that can replicate in invertebrate and vertebrate system.

8) The following paper is often overlooked in the CCHFV literature.:

Host-passage-induced phenotypic changes in crimean-congo haemorrhagic fever virus Gonzalez et al., 1995. If possible, it would be good to cite the paper in the beginning of the Discussion.

9) The authors refer to no "widely" approved vaccines for CCHF. Are there any approved vaccines? The word widely implies that there are vaccines that have limited distribution.

---

## [Author Response]

Revisions:1) One critique of the study is the lack of data detailing the clinical disease of MA-CCHFV itself. While weight loss is a useful metric in this model, I would have liked to have seen an evaluation of clinical signs throughout the challenge period, i.e., clinical scores at each day, which would help the reader better understand any inherent variability in the model. For example, and it is touched upon in the results, the large error bars in Figure 1B indicate that the female mice infected with MA-EBOV have a much more variable disease course than male mice – a representation of clinical scores would help with the reporting of this data. It may also be useful to represent individual animals in these weight loss graphs rather than use the mean, again for clarity. A clearer understanding of the expected disease course and clinical signs in a MA-CCHFV infection from challenge to recovery is would benefit any future research understanding the efficacy of potential antiviral compounds or vaccines.

Unfortunately, we did not collect clinical scores during these studies and thus the requested data does not exist. However, our subjective opinion is that weight loss correlated well with overt clinical signs of disease such as ruffled fur, hunched posture and lethargy. In addition to weight loss, we measured viral loads, inflammatory cytokines and liver enzymes that all correlated with disease severity. These objective parameters could be used as read outs for the efficacy of vaccines and antivirals.

Although the error bars in Figure 1B are larger, cumulatively over the course of the studies we have observed consistently less disease in female mice although on occasion we do have female mice that have disease similar to male mice. We have provided the data as individual mice in Author response image 1 however, we believe these graphs make discerning the groups difficult so we have presented them as mean plus standard deviation in the main text figures.

**Author response image 1. sa2fig1:** 

2) Why were the passaged viruses not plaque -picked, and can the authors speculate on how the viral swarm/quasi species is influenced by the passaging? The authors comment on the non-synonymous mutations but not really how the SNP change from passage to passage.

We avoided plaque-picking virus as we wished to avoid any growth of virus in tissue culture until final generation of stocks. We had concerns that passaging in tissue culture may select against mouse-adapted variants. For example, a mouse-adapted variant may plaque poorly or not grow at all in human-cell based tissue culture and thus would be “lost” in this approach. We have further specified in the text that liver tissue was passed mouse to mouse without isolation or purification. However, we do agree that the viral swarm may be important and cannot claim that our passaging scheme is the only one that would have resulted in similar mouse adaptation. We have not done formal evaluations of the quasi-species when virus is grown in mouse liver tissue versus tissue culture. In addition, Table 1 does list the prevalence of identified mutations at passage 4, 9 and 11. By passage 4 all mutations except synonymous M A1502G are present at >80% of reads. M A1502G increased in frequency during passages.

3) Results first paragraph: can the authors include the onset of disease for each passage in the supplementary information?

We have included when we collected the sample in updated Figure 1—figure supplement 1 and the weight loss of each passage. Weight loss in these mice correlates with onset of other signs of disease. For this data it is important to consider that we did not weigh the piece of liver collected nor titer the amount of CCHFV in the homogenized sample and therefore inoculating virus dose at each passage is not uniform. It is also limited in that only one mouse was inoculated with each tissue sample at each passage. Thus, we caution drawing conclusions on the virulence of any particular passage based on this data alone. Importantly, we omitted to specify that following the first passage of CCHFV, CCHFV in the blood of the passage 1 mouse was transferred to the naïve passage 2 mouse. Thereafter liver tissue was used for all passages. We have updated the text accordingly.

4) Was there a reason than an intermediate dose (1,000 TCID50s) of MA-CCHFV was used when infecting multiple mouse strains rather than the 10,000 TCID50s used elsewhere?

While no inverse correlation between disease severity and virus dose was observed in B6 mice, we still had some concern that this may not hold true for other mouse strains. For example, with MA-EBOV an inverse correlation was seen in B6 but not BALBc mice (Haddock et al., 2018). To minimize the number of mice used, rather than perform dose finding in multiple mouse trains we picked 1000 TCID50 as it was a middle ground between doses we knew to reliably cause disease in B6 mice. For further model development in any of these strains it may be important to perform similar dose finding studies.

5) While MA-CCHFV was shown to be lethal in the 3wk old mice, do we have comparative data for for wt Hoti?

We have not performed similar studies with CCHFV strain Hoti. We expect that infection of young WT mice with parental Hoti would not result in lethal disease as these younger mice still possess an intact, if not fully developed, type I IFN response. However, it is possible that young WT mice may show greater disease or viral replication with parental Hoti compared to adult mice.

6) The authors state that the clinical presentation was milder in females. I might have overlooked this in the manuscript, but is there a statistical side-by-side comparison of males vs. females in their different clinical parameters (weight loss, disease severity/scoring)? Figure 1?

We agree that this statistical comparison is important and have included it in Figure 1—figure supplement 2. Male mice had significantly greater weight loss than female mice on days 5, 6, 7 and 8 PI in this study. Similar findings were found in Figure 6 and 7 although since the major conclusions of those figures are comparison between WT and KO strains, we have not included the male and female WT statistical comparison in those figures.

7) The authors state "NSm is dispensable for virus replication (Gerrard et al., 2007)". This is true for mammalian systems, howver, not in mosquitoes, where it determines vector competence. Han et al., 2016 demonstrated a non-synonymous amino acid change in NSm after the mouse-passaged CCHFV strain IbAr 10200 was passaged back in ticks. This could be thought of a “reversion” back to a virus that can replicate in invertebrate and vertebrate system.

We agree and have added text to reflect this finding.

8) The following paper is often overlooked in the CCHFV literature.:Host-passage-induced phenotypic changes in crimean-congo haemorrhagic fever virus Gonzalez et al., 1995. If possible, it would be good to cite the paper in the beginning of the Discussion.

We have added this reference and expanded the Discussion on host-selective pressures for CCHFV.

9) The authors refer to no "widely" approved vaccines for CCHF. Are there any approved vaccines? The word widely implies that there are vaccines that have limited distribution.

There is a Bulgarian vaccine using inactivated CCHFV grown in mouse brains that is approved for use in that country. Although this vaccine appears protective it is unlikely this vaccine would ever achieve wider distribution due to safety concerns inherent to how it is manufactured.